# Revealing key regulators of neutrophil function during inflammation by re-analysing single-cell RNA-seq

**Zhichao Ai** [ORCID] *

China Innovation Center of Roche (CICoR), Roche R&D Center (China) Ltd, Shanghai, China

* eric.ai@roche.com

**Citation:** Ai Z (2022) Revealing key regulators of neutrophil function during inflammation by re-analysing single-cell RNA-seq. PLoS ONE 17(10): e0276460. https://doi.org/10.1371/journal.pone.0276460

**Data Availability Statement:** Single Cell RNA Sequencing Raw data were obtained from the GEO database (GSE137540, GSE165276, GSE120409, and GSE158055).

## Abstract

Excessive neutrophil infiltration and dysfunction contribute to the progression and severity of hyper-inflammatory syndrome, such as in severe COVID19. In the current study, we re-analysed published scRNA-seq datasets of mouse and human neutrophils to classify and compare the transcriptional regulatory networks underlying neutrophil differentiation and inflammatory responses. Distinct sets of TF modules regulate neutrophil maturation, function, and inflammatory responses under the steady state and inflammatory conditions. In COVID19 patients, neutrophil activation was associated with the selective activation of inflammation-specific TF modules. SARS-CoV-2 RNA-positive neutrophils showed a higher expression of type I interferon response TF IRF7. Furthermore, IRF7 expression was abundant in neutrophils from severe patients in progression stage. Neutrophil-mediated inflammatory responses positively correlate with the expressional level of IRF7. Based on these results, we suggest that differential activation of activation-related TFs, such as IRF7 mediate neutrophil inflammatory responses during inflammation.

## Introduction

Neutrophils are important effector cells in innate immunity, possessing a wide range of effector functions, including reactive oxygen species (ROS) production, phagocytosis and chemotaxis. These properties enable neutrophils to rapidly respond to stimulation and orchestrate protective immunity [1, 2]. Currently, the transcriptional regulatory networks underlying neutrophil activation and function remain largely unexplored. Of note, there is partial understanding on transcription factors (TFs) that module as key regulators that control neutrophil development and inflammatory responses. Neutrophils undergo tightly controlled genomic and transcriptional changes while transitioning between bone marrow blood, and tissue sites. However, the molecular mechanisms underlying neutrophil active transcriptional remodeling during inflammation remain to be elucidated.

Recent data from single cell RNA sequencing (scRNA-seq) studies have provided important insights into the transcriptional regulatory networks underlining neutrophil heterogeneity under steady-state as well as inflammatory conditions [3–6]. These published scRNA-seq

**Funding:** The authors received no specific funding for this work.

**Competing interests:** The authors have declared that no competing interests exist.

studies have sampled neutrophils from bone marrow, blood, tissues, and differentially defined neutrophil subpopulations by distinct molecular signatures. However, the previously published analysis of the transcriptional regulatory networks behind neutrophil heterogeneity was limited. Therefore, we re-analysed published scRNA-seq datasets with a focus on the TF modules that underline neutrophil development and function. Investigating the repertoires of neutrophil-specific TFs may lead to multiple therapeutic strategies tailored to neutrophil-mediated inflammatory diseases, such as chronic inflammatory diseases and severe COVID19 infection.

## Results

### Neutrophil developmental trajectories

A number of neutrophil subpopulations have previously been identified through scRNA-seq analysis of neutrophils under physiological and various inflammatory conditions [4–7]. To obtain a harmonised neutrophil transcriptional dictating neutrophil differentiation and function, we integrated the scRNA-seq data of mouse neutrophils from bone marrow (BM) and blood, and projected into two dimensions with Uniform Manifold Approximation and Projection (UMAP) based on their transcriptomic profiles [8]. Unbiased, graph-based clustering identified four major neutrophil C1-4 subpopulations (**Fig 1A**). The maturation score of each differentiating neutrophil population, based on the expression of genes related to neutrophil differentiation [9], suggests the increasing maturity from C1 to C4 subpopulations (**Fig 1B**). Hematopoietic stem and progenitor cells (HSPCs), multi-lineage (Multi-Lin) cells, and bi-potential monocytic-granulocytic (IG2) cells, such as proNeu and preNeu. overlapped with G0-2 cells in the same defined cluster, while Ly6G-expressing P1-4 cells displayed overlapping clusters with immature and mature cells (G3-G5 cells) (**Fig 1C**), indicating there was little batch effect between samples.

To investigate the inter-interrelationship between scRNA-seq-defined neutrophil subpopulation, we conducted hierarchical clustering. Consistent with UMAP clustering, early neutrophil precursors, including HSPCs, Multi-Lin, IG2 cells, proNeu, and preNeu were closely associated with G0-2 and P1 cells, while G3-4 cells were more close to P3 and immature neutrophils. Neutrophils in the blood (G5 cells) were clustered closely with P4 cells (**Fig 1D**), representing neutrophils with the highest maturity. Finally, we examined the expression of various neutrophil granules in scRNA-seq-defined neutrophil subpopulations (**Fig 1E**). As expected, primary granules were highly expressed in cells under early stage of differentiation, such as IG2, G0-1 cells, proNeu and preNeu. In comparison, neutrophils with intermediate maturity, such as preNeu, G2-3 cells, P1-3 cells and immature neutrophils, remarkably upregulated secondary granules. whereas terminally differentiated neutrophils (P4 and G4-5 cells) highly expressed tertiary granules, consistent with the patterns of granule expression throughout neutrophil development [9].

### Transcriptional factor dynamics during neutrophil differentiation

We next investigated the transcriptional regulatory dynamics during neutrophil differentiation, because tightly controlled transcriptional programmes regulate neutrophil cell decision and function [10]. Unsupervised hierarchical clustering of the differentially expressed TFs across differentiation stages and identified three types of TF clusters (**Fig 2A**). TFs downregulated with maturation included previously reported differentiation factors *Cebpa*, *Cebpz* and *Gata2*, which are necessary for granulopoiesis initiation [10, 11], and some new regulons, such as *Bclaf1* and *Hcfc1*. TFs transiently upregulated in the intermediate differentiation stage included terminal differentiation factor *Cebpe* [9] and new regulon *Ets1*. Among the TFs

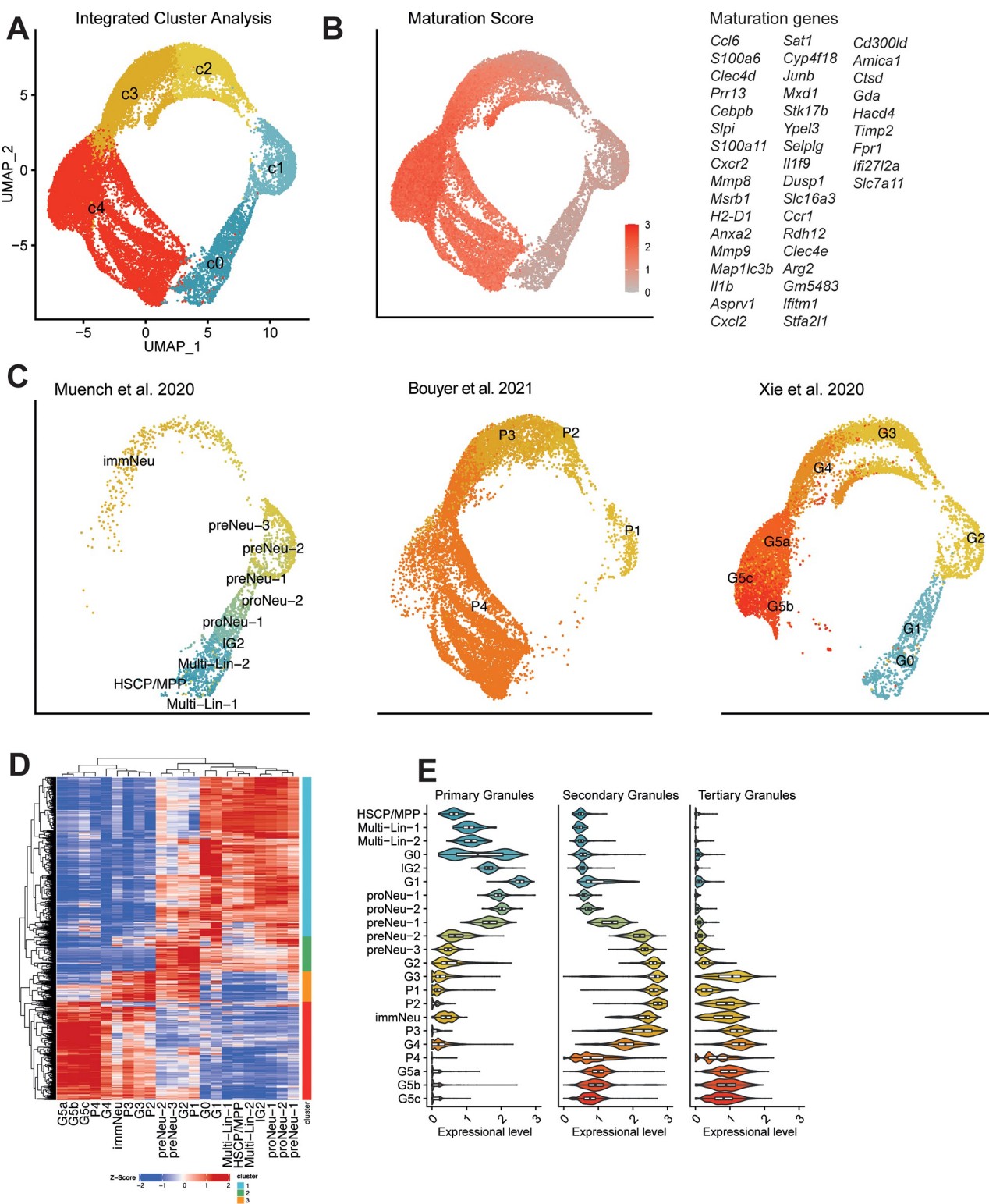

**Fig 1. Integrated scRNA-seq analysis of mouse neutrophils under differentiation.** (A) The Uniform Manifold Approximation and Projection (UMAP) presentation of 4 major cell clusters in mouse neutrophils from bone marrow and blood. (B) The UMAP map showing the maturation score of neutrophils from bone marrow and blood. (C) The UMAP presentation of cell clusters divided by the study of scRNA-seq neutrophils. (D) Unsupervised hierarchical clustering of all differentially expressed genes (|log2(fold change|>1) and padj < 0.01), based on Manhattan distances using the Ward method. Data are presented as heatmap normalised to the minimum and maximum of each row. (E) Violin plot of maturation scores for neutrophil primary, secondary and tertiary granules in each scRNA-seq-defined cluster.

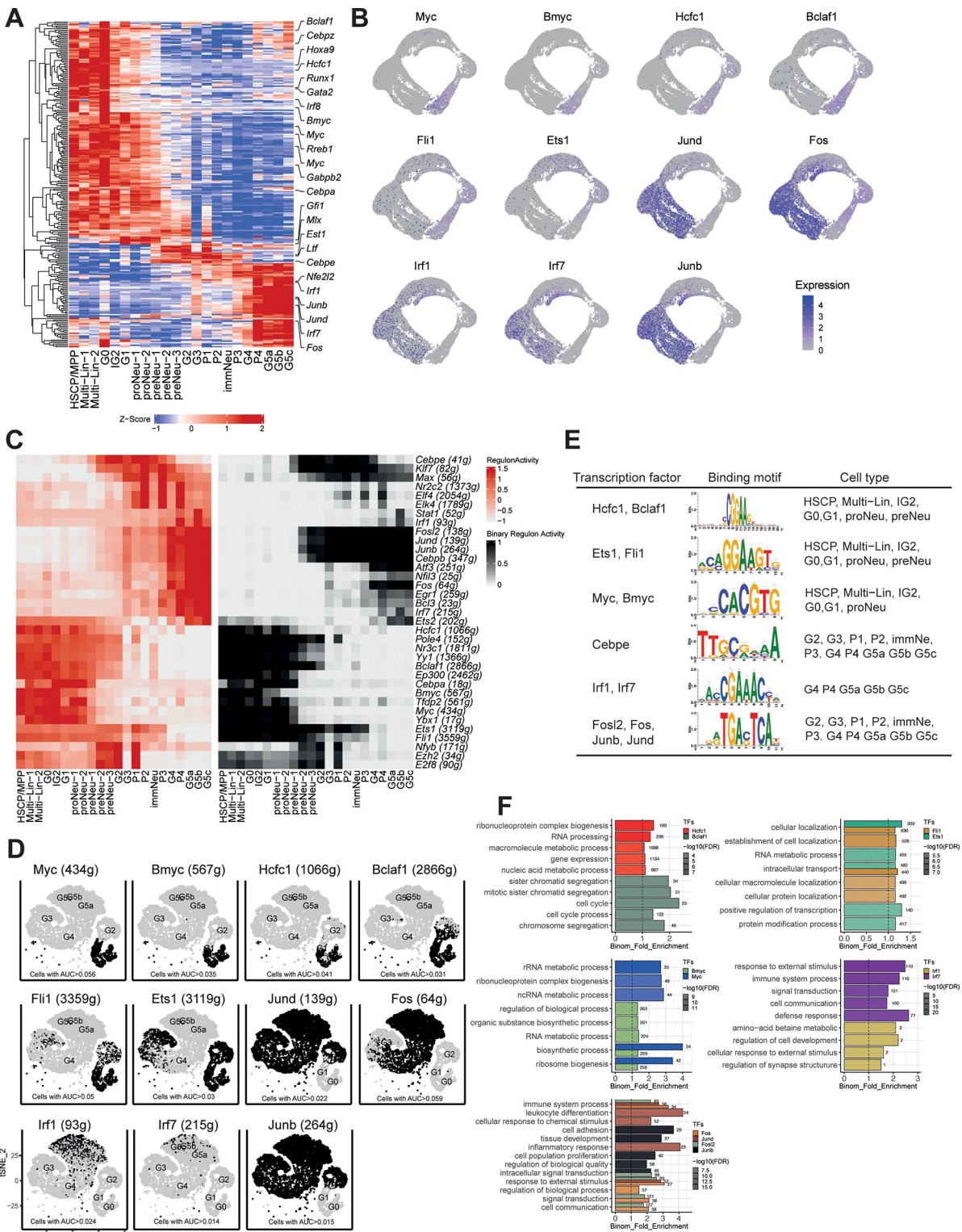

**Fig 2. Transcription factor networks of neutrophils along differentiation trajectories.** (A) Heatmap showing the expression of TFs across scRNA-seq-defined neutrophil subpopulations. (B) Expression of differentiation-related TFs. (C) SCENIC results for neutrophil transcriptional regulatory networks. Heatmap showing the expressed and active transcription factors in each cell class. The AUCell and BINary regulon activity of the transcription factors in each cell class are indicated as red (highly active) and white (lowly active), and black (active) and white (inactive), respectively. (D) The t-SNE presentation of key regulons in each scRNA-seq-defined neutrophil subpopulation.

(E) Signature regulons (rows) are clustered according to the binding motifs and cell types in which they are active. (F) The gene ontology analysis of regulated genes targeted by signature TFs across differentiation stages.

progressively upregulated during terminal differentiation are inflammatory regulons like *JunB* and *Nfe2l2*, as well as uncharacterised TFs, like *Jund*, *Fos* (**Fig 2B**). These results are consistent with previous studies supporting stage-specific transcriptional regulatory networks guiding neutrophil development [12, 13].

To identify TFs with high transcriptional activity during neutrophil differentiation, we applied single-cell regulatory network inference and clustering (SCENIC), a method that evaluates the activity of the gene expression network within each individual cell [14]. In the current study, TFs that are active in more than 50% of cells within a particular cell class are retained and active TFs are selected by both the AUCell and BINary algorithms (**Fig 2C**). Strikingly, this analysis identified 35 active TFs that regulate 6774 genes across neutrophil subpopulations across different samples (**S1 Table**). From each cluster of TFs, we identified some representative regulons and related cell types through their AUCell and BINary activities. When mapping the regulon activity of each TF onto the tSNE map, we found that each TF occupies a distinct region and all highlighted regions show complementary patterns (**Fig 2C & 2D**).

We next determined the regulatory events impacted by each TFs. During the progression of maturation, it was found that a dramatic decrease of biosynthetic and metabolic transcription activities, such as *Hcfc1*, *Bclaf1*, *Ets1*, *Fli1*, *Myc*, and *Bmyc*, and their target genes were enriched for various cellular and metabolic processes, including RNA processing, cell cycle, biosynthesis and transcriptional regulation (**Fig 2E & 2F**). In contrast, neutrophil transition toward mature stages is accompanied with elevated transcriptional activities of TFs, such as *Irf1*, *Irf7 Fosl2*, *Fos*, *Junb*, *Jund*, to their target genes, which are more related to pathogen clearance and bactericidal activities (**Fig 2E & 2F**). By utilising the bulk RNA-seq results reported by Evrard *et al.* [9], consistently we found the expression of *Irf1*, *Irf7*, *Fos*, *Junb*, *Jund* increased with neutrophil maturation (**S1A Fig**).

## Inflammation-specific TFs during bacterial infection

We then assessed the gene expression difference between bacterial infection and non-inflammatory condition (**Fig 3A**). We first examined the expression of genes known to be involved in neutrophil development and activation. Genes that increase the extent of neutrophil differentiation were increased in expression in neutrophil precursor G0-1 cells, whereas neutrophil activation genes were more upregulated in G3-4 cells approaching terminal differentiation (**Fig 3B**). To further dissect the regulatory difference induced by bacterial infection, we conducted the differential TF expression and SCENIC analysis to compare the TF expression and regulon activities in each neutrophil subpopulation (**Fig 3C**). This identified 8 active inflammation-specific TFs, which were significantly upregulated and activated in one of neutrophil subpopulations in response to infection. For instance, *Jund*, *Fos*, *Pole3*, *Pole4* were enriched and activated in proliferating G0-1 cells, whereas *Irf7*, *Nfe2l2*, *Bcl3* were more expressed and activated in relatively mature G4-5 cells (**Fig 3D**).

Mapping the region activity of those TF onto the t-SNE map, we were able to identify distinct transcriptional regulatory signatures in each neutrophil subpopulation (**Fig 3E**). In the gene ontology analysis, the target genes of *Fos*, *JunD*, *Nfe2* were involved in regulating cellular and RNA metabolic processes, suggesting a metabolic adaptation in early progenitor cells (**Fig 3F**). A set of *Pole3*, *Pole4* target genes were enriched in chromosome segregation and cell division, consistent with accelerated neutrophil generation during inflammation (**Fig 3F**). Additionally, genes targeted by *Irf7*, *Nfe2l2*, *Bcl3* preferentially participated in immune system

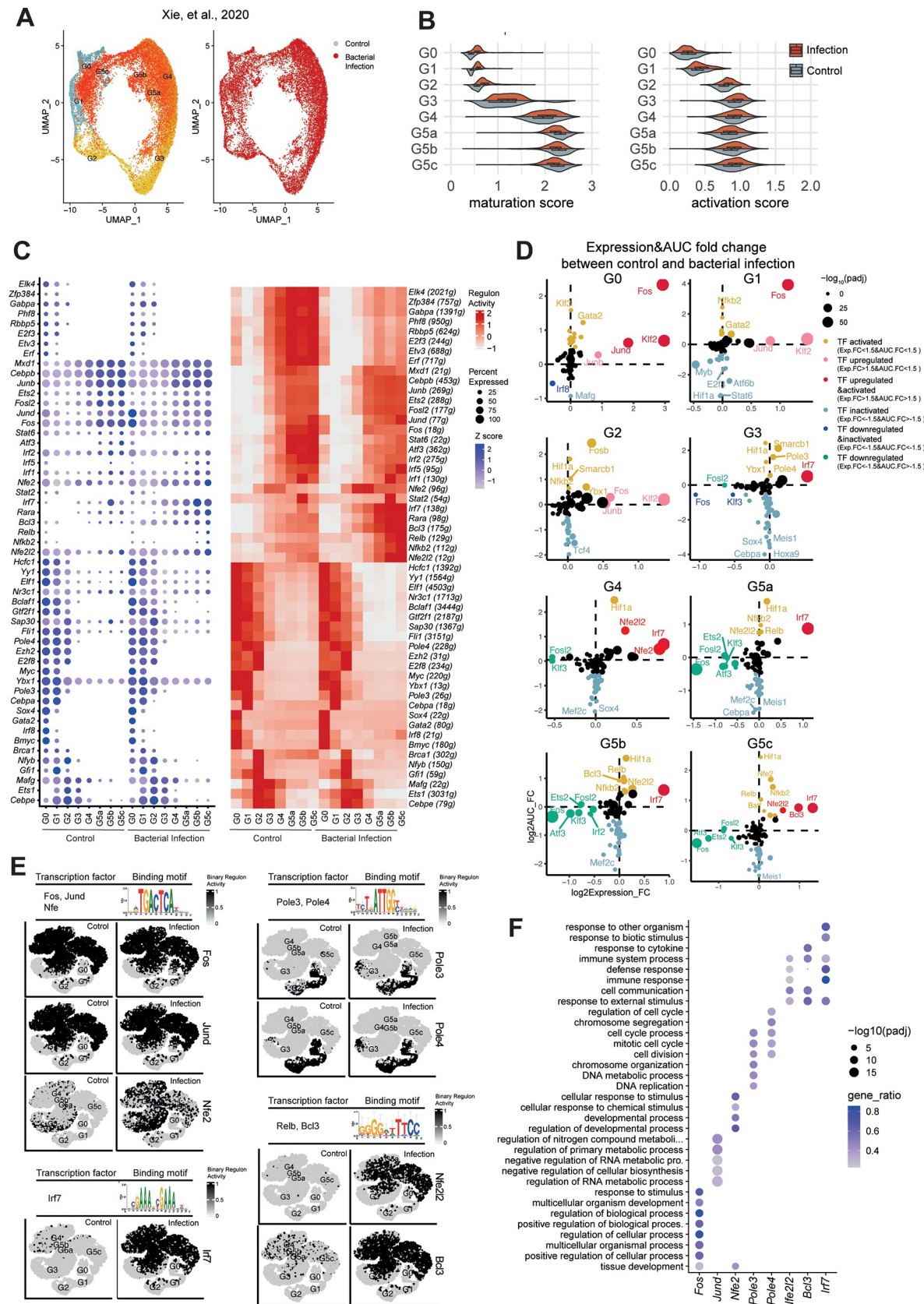

**Fig 3. Inflammation-specific transcription factor networks in neutrophils.** (A) The (UMAP) presentation of scRNA-seq-defined neutrophils (progressively maturing Go to G5 clusters) under bacterial infection and control. (B) the activation and maturation score of neutrophil subpopulations. (C) TF networks of neutrophil subpopulations under bacterial infection and control. (Left) Expression of signature TFs in each cell class (Go to G5 clusters). (Right) the AUCell regulon activity of signature TFs in each cell class (Go to G5 clusters) are indicated as red (highly expressed) and white (lowly expressed). (D) Intersection of infection-vs-control changes in signature TFs expression and AUCell regulon activity: TFs which are both upregulated and active are highlighted in red, TFs which are both downregulated and inactive are highlighted in blue. Only-upregulated TFs highlighted in pink. Only-active TFs highlighted in orange. (E) The t-SNE presentation of key regulons in each cell class. (F) Dot plot indicating the gene ontology analysis of regulated genes targeted by signatures TFs in each cell class.

process and response to external stimulus, supporting augmented neutrophil functionality during inflammation (**Fig 3F**). Based on the bulk RNA-seq analysing transiting neutrophils during inflammation [15], it was found that enhanced expression of *JunD*, *Fos*, *Elf2*, *Irf5*, *Irf7* were observed during neutrophil transition from the bone marrow, and to the inflammatory sites (air pouch membrane (MEM) and exudate (AP) (S1B Fig)), consistent with our current study.

## Transcriptional regulator networks in COVID19 infection

To investigate whether similar transcriptional networks exist in human disease, we analysed published scRNA-seq of the lung and peripheral blood from coronavirus disease 2019 (COVID-19) patients, in which the presences of distinct neutrophil subsets associated with COVID19 severity and stage were reported [16]. We followed their Seurat-pipeline to stream-line cell clustering along major neutrophil subpopulations, including Neu_c1−IL1B, Neu_c2−CXCR4$^{low}$, Neu_c3CST7, Neu_c4−RSAD2, Neu_c5−GSTP1$^{high}$OASL$^{low}$, and Neu_c6−FGF23 cells (**Fig 4A**). To dissect neutrophil heterogeneity, we examined the differentially expressed genes (DEGs) analysis and found that there were substantial differential genes in each neutrophil subpopulation (**Fig 4B**, **S2 Table**). In the gene ontology analysis of DEGs, activation-related genes, and differentiation-related genes were expectedly expressed by Neu_c1−IL1B, whereas differentiation-associated genes were highly expressed by Neu_c2−CXCR4$^{low}$. Additionally, Neu_c3CST7 started to upregulate gene central to antigen presentation and pattern receptor recognition. Neu_c4−RSAD2, and Neu_c5−GSTP1$^{high}$OASL$^{low}$ cells highly expressed genes associated with type I interferon responses and neutrophil inflammatory responses, respectively. Interestingly, genes related to chromatin remodeling were highly expressed by Neu_c6−FGF23 (**Fig 4C**).

We then measured the expression score of neutrophil subpopulations based on genes related to neutrophil activation, type I interferon response, antigen presentation, cytokine and chemokine production. It was found that Neu c2−CXCR4$^{low}$ cells, which were less responsive under COVID19 infection, where as Neu_c4−RSAD2, and Neu_c5−GSTP1$^{high}$OASL$^{low}$ were specialised in type I interferon responses and inflammatory responses (**Fig 4D**). We then assessed how COVID19 infection affected the transcriptional regulatory networks across neutrophil subpopulations at transcription regulatory level (**Fig 4E**). Changes in TFs networks, such as JUND, FOS, ELF2, AHCTF1 were more prominent in Neu_c1−IL1B, Neu c2−CXCR4$^{low}$, and Neu_c6−FGF23, whereas IRF5, STAT1/2, and IRF7 were commonly active in Neu_c4−RSAD2, and Neu_c5−GSTP1$^{high}$OASL$^{low}$ (**Fig 4E**).

We reasoned that the systemic inflammatory storm might be associated with the transcriptional regulatory activation in neutrophils, since dysregulated mature neutrophils have been frequently observed in severe COVID19 patients [17]. Then, we examined the expression pattern of signature TFs in neutrophils from mild and severe COVID19 patients, and found that IRF7 was significantly upregulated in neutrophils from severe COVID patients under progression stage (**Fig 4F**). However, no increase of *IRF7* was observed in mild COVID19 patients

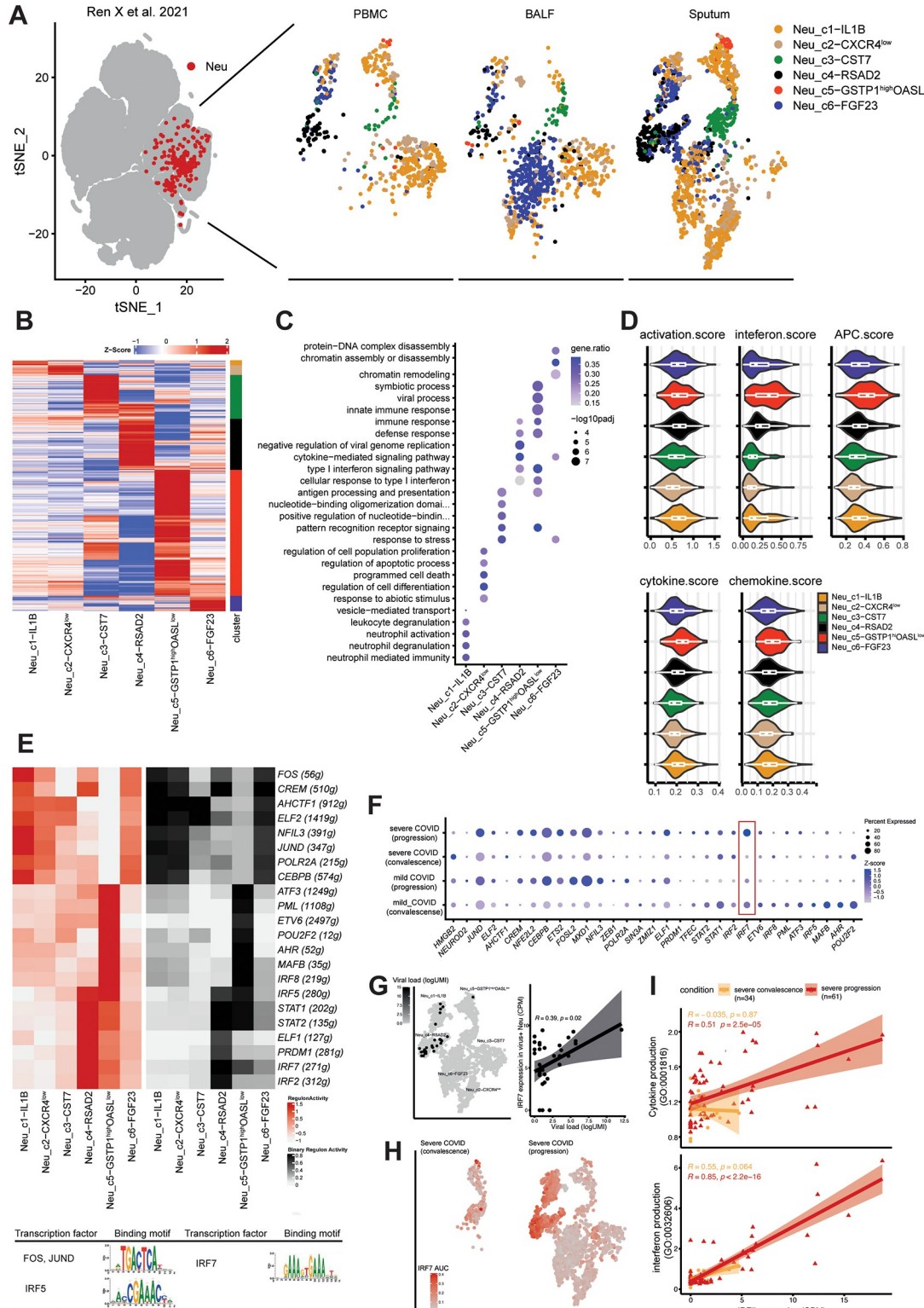

**Fig 4. Neutrophil transcriptional regulatory networks in COVID19 infection.** (A) The t-SNE presentation showing major neutrophil subpopulations from the peripheral blood, bronchoalveolar lavage fluid, and sputum from COVID19 patients. (B) Hierarchical clustering of differentially expressed genes (DEGs) (padj < 0.01, |log2FC| >1), based on Manhattan distances using the Ward method. Data are presented as heatmap normalised to the minimum and maximum of each row. (C) The gene ontology analysis of DEGs by each neutrophil subpopulation indicated in (B). (D) the average expression levels of genes central

to neutrophil activation, interferon I response, antigen presentation, cytokine and chemokine production in each neutrophil subpopulation in COVID19 patients. (E) Expression and AUCell regulon activity of signature TFs in neutrophil subpopulation, indicated as red (highly expressed) and white (lowly expressed), and black (active) and white (inactive). (F) Dot plot indicating the expression of signature TFs in neutrophils from mild and severe patients under convalescence and progression stage. (G) Viral load in COVID19-related neutrophils. (Left) the t-SNE map indicating the level of viral load in neutrophils. (Right) Correlation of IRF7 expression to the level of viral load in virus-positive neutrophils. The correlations were tested using the one-tailed Spearman's test. (H) The t-SNE map showing the expression of *IRF7* expression in severe patients under convalescence and progression stage. (I) The correlation between *IRF7* expression and the expressional levels of genes associated with cytokine production and type I interferon response in neutrophils from several patients under convalescence and progression stage. One-tailed Spearman's test was used for the correlation analyses.

under progression, supporting the existence of IRF7 hyper activation in neutrophils from progressing severe COVID19 patients.

The presence of SARS-CoV-2 RNA in immune cells seemed to be associated with additional transcriptomic changes [16]. For neutrophils, SARS-CoV-2 RNA were mostly detected in Neu_c4–RSAD2, and Neu_c5–GSTP1$^{high}$OASL$^{low}$ and the viral load positively correlated with the expressional level of IRF7 in neutrophils (**Fig 4G**). Consistently, higher IRF7 expression was observed in severe patients under progression stage (**Fig 4H**). To determine the functional difference of IRF7 activation between mild and severe patients, we measured the cytokine production and type I interferon score of neutrophils between severe patients under convalescence and progression stage. As a result, neutrophil IRF7 expression was positively correlated with the average expression of genes central to cytokine production and type I interferon responses (**Fig 4I**). Therefore, IRF7 may also regulate cytokine and type I interferon responses in human neutrophils.

Excessive inflammation has been a risk factor for infection [18], highlighting an unmet need for IRF7 inhibition for reducing excessive inflammation. The therapeutic value of IRF7 inhibition has been demonstrated by the experiment that mice treated with IRF7 inhibitors were protected against bacterial infection without unwanted renal tissue damage (**S3 Table**).

## Discussion

It has been increasingly recognised that neutrophils are transcriptionally active cells and undergo chromatin and transcriptional remodeling in adaption into various tissues and microenvironment [19, 20]. In the current study, we compared the transcriptional regulatory networks during neutrophil transition under distinct developmental stages, activation status and conditions of COVID19 infection, and demonstrated that distinct sets of TF modules regulate neutrophil transition between consecutive developmental stages. Moreover, we have identified previously unappreciated activation of inflammation-specific TFs as a result of the neutrophil activation. Further validation of signature TFs, using COVID19 immune cells scRNA-seq dataset, demonstrated the potential role of IRF7 in neutrophil-mediated inflammatory responses during COVID19 infection.

Previous studies have shown that tightly-controlled transcriptional programmes regulate neutrophil differentiation [21, 22]. This is likely to reflect on the transcriptome and transcription differences that occur across the differentiation stages. Indeed, by reanalysing the transcriptional regulatory alterations of scRNA-seq-defined neutrophil sub-populations from different samples, we discovered the involvement of distinct cluster of TFs during the developmental transition. For instance, *Myc*, *Cebpe* and *Junb* are differentially expressed and activated in differentiating neutrophils, each associated with a set of target genes central to neutrophil differentiation. In addition to previously reported differentiation TFs, we identified several uncharacterised regulons, such as *Fos*, *Jund*, *Ets1*, *Hcfc1*, *Bclaf1*, which likely regulate the

expression of neutrophil-specific genes. However, subsequent functional validation of these TFs are required.

One interesting result of our analysis is the identification of the inflammation-specific TF modules, as defined by the increased transcription and regulon activities in inflamed neutrophils. These transcriptomic and regulatory differences reflect the importance of change induced by inflammation. Emulating earlier studies [4, 6, 23], we found a transcriptional switch from cellular-and-metabolic programmes into inflammatory-response programmes, perhaps driven by the activation of a set of TFs, such as *Fos*, *Jund*, *Pole3*, *Pole4*, *Irf7*, *Nfe2l2* and *Bcl3*, and the upregulation of their regulated genes that promote neutrophil maturation. Transcriptional regulatory differences may provide a reason for inflammation-induced augmented neutrophil generation.

Neutrophil hyperactivation is a feature of severe COVID19 patients, and therapeutic targeting of neutrophils has been proposed by clinicians to reduce the severity of COVID-19 infection [24]. Despite the importance of TFs in mediating neutrophil activation, the role and contribution of activation-related TFs in COVID19 infection remain unexplored. Thus, we re-analysed the transcriptional regulatory differences between neutrophil subpopulations from mild and severe patients, and found that elevated transcription and regulon activity of TFs, such as *IRF5*, *STAT1/2*, and *IRF7*, and their regulated genes in neutrophil subpopulations with enhanced effector mechanisms. Interestingly, severe patients under progression showed increased level of *IRF7* expression in comparison to convalescent patients. We also observed a positive correlation of *IRF7* expression to the load of SARS-CoV-2 RNA in virus-positive neutrophils. Our analysis supported that IRF7 expression was positively correlated with the expressional levels of genes central to cytokine production and type I interferon responses, suggesting that IRF7 activation in neutrophils mediates neutrophil cytokine production and type I production during COVID19 infection. Thusly, IRF7 signaling blockage might be effective in reducing neutrophilic inflammation during COVID19 infection. Although the finding that activation-related TFs like *IRF7* may be relevant with neutrophil hyperactivation is novel, it still should be functionally validated both *in vitro* and *in vivo* in future studies.

## Limitations of study

In the current study, we re-analysed single-cell transcriptional heterogeneity of mouse and human neutrophils under steady and inflammatory conditions and identified distinct sets of signature TFs that potentially mediate neutrophil development, activation, and inflammatory responses. Future experimental validation of the functional role of these signature TFs would be valuable to characterise the transcriptional regulatory dynamics underlining neutrophil phenotype and function in tissues as well as various inflammatory conditions, such as autoimmune diseases and cancers.

## Data and code availability

The scRNA-seq data used in the current study is publicly available with the GEO database.

scRNA-seq of differentiating and mature neutrophils under homeostasis and bacterial infection is from https://www.ncbi.nlm.nih.gov/geo/query/acc.cgi?acc=GSE137540

scRNA-seq of CD11b+Ly6G+ murine neutrophils from healthy tissues and experimental inflammation is from https://www.ncbi.nlm.nih.gov/geo/query/acc.cgi?acc=GSE165276

scRNA-seq of murine neutrophils under specification and commitment is from https://www.ncbi.nlm.nih.gov/geo/query/acc.cgi?acc=GSE120409

scRNA-seq of COVID-19 patients with differential disease severity and various tissue types is from https://www.ncbi.nlm.nih.gov/geo/query/acc.cgi?acc=GSE158055

## scRNA-seq data: QC and defining major cell types

Single-cell RNA-seq counts were obtained from the GEO database as the raw expression matrix of unique molecular identifier counts (UMI counts; indicative of the number of unique RNA molecules detected) for each gene in each cell group. UMI counts from each dataset were filtered on the following criteria: cells with > 1000 UMI counts, > 200 features, < 7500 features and < 25% mitochondrial gene expression, then normalised and log-transformed using the LogNormalize function in the Seurat package. We integrated all the datasets using the 'scTransform' package and 'harmony' package for batch correction and scRNA-seq data integration. It was then followed by a combined principal component analysis (PCA), computing 50 principal components, of which the first 20 (based on an elbow plot) were used to compute the Uniform Manifold Approximation and Projection (UMAP) analysis for dimensionality reduction. Next, we clustered cells, exploring a range of different resolution settings of k = 20/100/500 and resolution of 0.8. Cell clusters were then annotated following the annotation of the original researches.

## Identification of DEGs

To identify the differentially expressed genes (DEGs), we ran a modified version of the Findmarkers by applying the average expression of all the genes across each cell types and setting the logfc.threshold = 0.5(corresponding to a fold-change of 1.5) and p_val_adj<0.05 to identify differentially expressed genes (DEGs). P value adjustment was performed using Bonferroni correction based on the total number of genes in the dataset. Gene Ontology analysis was performed by using the R package goTOP [25].

## Scoring of biological processes

For analysis, we calculated the score of biological and cellular processes by averaging the expression of gene signatures representing defined biological functions. The neutrophil activation, maturation and granule signatures signature was derived from [9]. Type I interferon production, antigen presentation, cytokine and chemokine production were derived from the GO term 'type I interferon production' (GO: 0032606), 'antigen processing and presentation' (GO: 0019882),'cytokine production' (GO: 0001816), 'chemokine production' (GO: 0032602). Maturation-related genes were summarized from the previous literature. Neutrophil maturation is a process of the committed proliferative neutrophil precursors differentiating into non-proliferative mature neutrophils with competent effector functions. Related to function, mature neutrophils express extracellular matrix enzyme Mmp8, pro-inflammatory cytokine and chemokine *Il1b*, *Ccl6*, chemokine receptor *Cxcr2*, and maturation related transcriptional factor (TF) *Cebpb*. Neutrophil activation score is a collection of genes related to neutrophil activation (GO:0042119). It reflects the functional behaviors of neutrophils in response to pro-inflammatory cytokines, soluble factors and other inflammatory stimuli. Upon bacterial infection, neutrophils enhance their capacity in phagocytosis, chemotaxis, cytokine production and other effector functions. For all the functional signatures listed, scores were calculated as the average normalized expression of corresponding genes.

## SCENIC analysis

SCENIC is a computational method to guide the identification of regulatory regulons by and their putative target genes characterised by enrichment of corresponding transcription factor-binding motifs in their regulatory regions [14]. Regulatory network analysis was performed using R package SCENIC (version 3.0.1) with default parameters and procedures. Output co-

expression modules were trimmed with cisTarget databases (mm10_refseq-r80_500bp_u-p_and_100bp_down_tss, mm9-tss-centered-10kb-7species and mm9-500bp-upstream-10species). The identified regulons were then scored to determine their activities in each cell, indicated in each section, and then visualised according to the recommended protocol. For visualisation, we calculated the average regulon activity (AUC) scores for each neutrophil cluster and selected the top regulons to plot as t-SNE using the SCENIC package and as a heatmap using the pheatmap package.

## Supporting information

**S1 Fig. Transcriptional change of neutrophil subpopulations.** (A) Heatmap showing row-scaled expression of the signature TFs in bulk RNA-seq analysis of neutrophil populations in the bone marrow and blood (Evrard et al. 2018, Immunity). (B) Heatmap showing the scaled expression of signature genes for each bone marrow neutrophil subpopulation from the bone marrow(BM), blood(BL), air pouch membrane(MEM) and exudate(AP), coloured by the average expression of each gene in each cluster scaled across all clusters (Tariq et al. 2022, Nature Immunology).
(TIF)

**S1 Table. The list of differentially expressed genes across each stage of neutrophil subpopulation under differentiation.**
(CSV)

**S2 Table. The list of differentially expressed genes across neutrophil subsets from COVID-19 patients and healthy donors.**
(CSV)

**S3 Table. The preclinical data on IRF7 inhibition for treating bacterial infection.**
(XLSX)

## Acknowledgments

The suggestions from Hong Shen and Chengcheng Liu from China Innovation Center of Roche (CICoR) were greatly appreciated for the current study.

## Author Contributions

**Data curation:** Zhichao Ai.

**Formal analysis:** Zhichao Ai.

**Investigation:** Zhichao Ai.

**Methodology:** Zhichao Ai.

**Visualization:** Zhichao Ai.

**Writing – original draft:** Zhichao Ai.

**Writing – review & editing:** Zhichao Ai.

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
