## [Decision Letter · Decision Letter 0]

3 Aug 2022

PONE-D-22-13750Revealing key regulators of neutrophil function during inflammation by re-analysing single-cell RNA-seqPLOS ONE

Dear Dr. Ai,

Thank you for submitting your manuscript to PLOS ONE. After careful consideration, we feel that it has merit but does not fully meet PLOS ONE’s publication criteria as it currently stands. Therefore, we invite you a revised version of the manuscript that addresses the points raised during the review process. Especially, what is the new fidnings from this re-analysis, validate gene expression profiles with bulk RNAseq data, and discuss potential clinical significance of targeting IRF7 for treatment of COVID-19 and/or COVID-19 inflammation.

We look forward to receiving your revised manuscript.

Kind regards,

You-Yang Zhao

Academic Editor

PLOS ONE

Journal Requirements:

"This study is of no competing interests."

e) Please provide an amended Funding Statement that declares *all* the funding or sources of support received during this specific study (whether external or internal to your organization) as detailed online in our guide for authors at http://journals.plos.org/plosone/s/submit-now.  

f) Please state what role the funders took in the study.  If any authors received a salary from any of your funders, please state which authors and which funder. If the funders had no role, please state: "The funders had no role in study design, data collection and analysis, decision to publish, or preparation of the manuscript." 

Please send your amended statements by return email; we will change the online submission form on your behalf.

Reviewers' comments:

Reviewer's Responses to Questions

**Comments to the Author**

1. Is the manuscript technically sound, and do the data support the conclusions?

Reviewer #1: Yes

Reviewer #2: Yes

2. Has the statistical analysis been performed appropriately and rigorously? 

Reviewer #1: N/A

Reviewer #2: Yes

3. Have the authors made all data underlying the findings in their manuscript fully available?

Reviewer #1: Yes

Reviewer #2: Yes

4. Is the manuscript presented in an intelligible fashion and written in standard English?

Reviewer #1: Yes

Reviewer #2: Yes

5. Review Comments to the Author

Reviewer #1: The current manuscript (PONE-D-22-13750) re-analyzed scRNA data from GEO and focused on the transcription factors. Some TFs were dig out to involve in neutrophil development, mature, activation under disease conditions such as infection. There were some concerns about the manuscript.

1. For expression assay, usually scRNA-seq is not enough depth. Thus could you find RNA bulk-seq data to support or validate the current finding?

2. In the current analysis, UMAP and t-SNE were used. Can you explain the reasons that different methods were used?

3. In DEG, can you first calculate the mean expression of clusters and then do analysis because FINDMARKER function would pick up genes from major variable features?

4. More details how the data were integrated should be provided in the methods. For example, in method about scRNA-seq analysis, scRNA-seq data from BALF from patients was indicated from four GEO documents. This was wrong.

5. The details how to score in method about biological processes should be indicated.

6. What was the original findings of these scRNA-seq data? What is the difference from re-analysis?

Reviewer #2: In this report, the author compared the transcriptional regulatory networks

of neutrophils at different developmental stages, activation status

and conditions of COVID19 infection. Activation of some TFs has been identified as a results

of neutrophil activation, and IRF7 expression was abundant in neutrophils from patients with COVID19 infection. Overall, this report demonstrated few novel interesting findings with very little conceptual advance. The only interesting observation is that IRF7 expression is higher in COVID19 patients, however the author has not explored further how IRF7 signaling might be useful for the treatment of COVID19.

6. PLOS authors have the option to publish the peer review history of their article (what does this mean?). If published, this will include your full peer review and any attached files.

Reviewer #1: No

Reviewer #2: No

---

## [Author Response · Author response to Decision Letter 0]

5 Sep 2022

Comments to the Author

Reviewer #1: The current manuscript (PONE-D-22-13750) re-analyzed scRNA data from GEO and focused on the transcription factors. Some TFs were dig out to involve in neutrophil development, mature, activation under disease conditions such as infection. There were some concerns about the manuscript.

1. For expression assay, usually scRNA-seq is not enough depth. Thus, could you find RNA bulk-seq data to support or validate the current finding?

We are grateful for the provided suggestion to re-analyze the published bulk-RNA-seq datasets to confirm the change of neutrophil specific transcription factors (TFs) during homeostasis and inflammation, which here we have now done.

We utilized two independent published neutrophil RNA-seq datasets (Evrard et al. 2018, Immunity; Tariq. 2022, Nat. Immunol) to examine the expression of the TFs identified and validated in our current study, and other members of their families, during neutrophil maturation at homeostasis. The expression of Runx1 was consistently highest at the earliest stages of neutrophil differentiation, whereas the expression of Irf1, Irf7, Fos, Junb, Jund increased with neutrophil maturation (Rebuttal FigR1A), consistent with what we have observed in our current study. Additionally, enhanced expression of JunD, Fos, Elf2, Irf5, Irf7 were observed during neutrophil transition from the bone marrow, and to the inflammatory sites (air pouch membrane (MEM) and exudate (AP) (Rebuttal FigR1B). The data now are added into the manuscript as Supplementary Figure1.

Therefore, we concluded that the list of TFs identified from our current analysis reflected chiefly on the transcriptional change of neutrophil maturation, activation under healthy and disease conditions.

Rebuttal Figure R1. Transcriptional change of neutrophil subpopulations. (A) Heatmap showing row-scaled expression of the signature TFs in bulk RNA-seq analysis of neutrophil populations in the bone marrow and blood (Evrard et al. 2018, Immunity). () Heatmap showing the scaled expression of signature genes for each bone marrow neutrophil subpopulation from the bone marrow(BM), blood(BL), air pouch membrane(MEM) and exudate(AP), coloured by the average expression of each gene in each cluster scaled across all clusters (Tariq et al. 2022, Nature Immunology).

2. In the current analysis, UMAP and t-SNE were used. Can you explain the reasons that different methods were used?

Many thanks for the provided comment, which leads to the better interpretation of our results. For UMAP, as the Figure1 demonstrated, UMAP-based trajectory analysis suggests the developmental relationship: (1) from Hematopoietic stem and progenitor cells ->, and (2) from retinal progenitor cells -> Early neutrophil precursors -> Immature and mature neutrophils cells. This branching closely captured the temporal order of cell types that arose along neutrophil generation and overall a more accurate presentation of neutrophil developmental trajectory. For t-SNE, it moves the high dimensional graph to a lower dimensional space points by points at a higher resolution. Both UMAP and t-SNE are non-linear, graph-based methods based on PCA analysis and used for dimensionality reduction. However, for the regulon network analysis executed by the SCENIC package (illustrated in Figure2&3), the tSNE-based visualization is the only available option to use in the package.

3. In DEG, can you first calculate the mean expression of clusters and then do analysis because FINDMARKER function would pick up genes from major variable features?

We thank the review for providing the comment that help us improve our analysis. We are well aware that the function FINDMARKERS from the Seurat package for examining the differential expression between groups of cells with a focus on major variable features, which could result in biased results. Therefore, for the differential expression analysis (specifically, Figure4 B&C), we extracted the average counts with the function AverageExpression(from Seurat) for each cell cluster, before deriving the differentially expressed genes for gene ontology enrichment/other downstream analysis, in agreement with what has been suggested by the reviewer.

4. More details how the data were integrated should be provided in the methods. For example, in method about scRNA-seq analysis, scRNA-seq data from BALF from patients was indicated from four GEO documents. This is wrong.

Very grateful for the reviewer for pointing out our mistake, which we have corrected in the method section and provided information that is more detailed on.

5. The details how to score in method about biological processes should be indicated.

Many thanks for pointing it out, and it is now indicated in the method section.

6. What was the original findings of these scRNA-seq data? What is the difference from re-analysis?

It is very good question for us especially when our current analysis stands on the shoulder of numerous giants (including but not limited to Evrad et al, Immunity 2018; Grassi, Cell Reports 2018; Adrover et al, Immunity 2019; Kwok et al, Immunity 2020; Xie et al, 2020; Ballesteros et al, Cell 2020), by whom several reference datasets have been generated. These datasets are of great importance to explore some potential regulators for neutrophil targeting therapy.

The main discoveries of our current analysis are: 1, Integrated analysis of scRNA-seq of neutrophils under during homeostasis and inflammatory conditions supported distinct sets of putative transcription factors associated with control of neutrophil maturation an.d inflammatory responses. 2, Re-analysis of COVID-19 scRNA-seq datasets with a focus on the immunological feature of neutrophils found the correlation of neutrophil-specific IRF7 expression to the patient’s disease severity. 3, The current study highlights IRF7 as a key regulator for modulating neutrophil function without causing excessive tissue damage. Currently, some IRF5 inhibitors has been in the preclinical stage for ensuring appropriate inflammation during infection.

 

Reviewer #2: In this report, the author compared the transcriptional regulatory networks

of neutrophils at different developmental stages, activation status

and conditions of COVID19 infection. Activation of some TFs has been identified as a result of neutrophil activation, and IRF7 expression was abundant in neutrophils from patients with COVID19 infection. Overall, this report demonstrated few novel interesting findings with very little conceptual advance. The only interesting observation is that IRF7 expression is higher in COVID19 patients; however, the author has not explored further how IRF7 signaling might be useful for the treatment of COVID19.

Many thanks for your comments, and our current study utilizing a list of publicly available single cell RNA-seq datasets to be further explored in identifying novel therapeutic targets for treating neutrophil-related diseases. Our findings offer a response to the classical but unresolved question of “good or bad” neutrophil function regulators and identify IRF7 as a key regulator for modulating neutrophil inflammatory response. We propose IRF7 inhibition as a therapeutic strategy to protect against excessive tissue damage and bacterial infection, as evidenced by the company Pharma SelectImmune Pharma, which is developing a siRNA therapy to inhibit interferon regulatory factor 7 (IRF7) and thus inhibit inflammation in infected tissue, reducing the risk of sepsis and renal infections(detailed below).

---

## [Decision Letter · Decision Letter 1]

7 Oct 2022

Revealing key regulators of neutrophil function during inflammation by re-analysing single-cell RNA-seq

PONE-D-22-13750R1

Dear Dr. Ai,

We’re pleased to inform you that your manuscript has been judged scientifically suitable for publication and will be formally accepted for publication once it meets all outstanding technical requirements.

Kind regards,

You-Yang Zhao

Academic Editor

PLOS ONE

Additional Editor Comments (optional):

Reviewers' comments:

Reviewer's Responses to Questions

**Comments to the Author**

1. If the authors have adequately addressed your comments raised in a previous round of review and you feel that this manuscript is now acceptable for publication, you may indicate that here to bypass the “Comments to the Author” section, enter your conflict of interest statement in the “Confidential to Editor” section, and submit your "Accept" recommendation.

Reviewer #1: All comments have been addressed

2. Is the manuscript technically sound, and do the data support the conclusions?

Reviewer #1: Yes

3. Has the statistical analysis been performed appropriately and rigorously? 

Reviewer #1: Yes

4. Have the authors made all data underlying the findings in their manuscript fully available?

Reviewer #1: Yes

5. Is the manuscript presented in an intelligible fashion and written in standard English?

Reviewer #1: Yes

6. Review Comments to the Author

Reviewer #1: All concerns were addressed properly. I think that the current MS is enough quality to publish on Plos One. Thus I suggested the MS should be acceptable.

7. PLOS authors have the option to publish the peer review history of their article (what does this mean?). If published, this will include your full peer review and any attached files.

Reviewer #1: No

---

## [Editor Report · Acceptance letter]

13 Oct 2022

PONE-D-22-13750R1 

Revealing key regulators of neutrophil function during inflammation by re-analysing single-cell RNA-seq 

Dear Dr. Ai:

I'm pleased to inform you that your manuscript has been deemed suitable for publication in PLOS ONE. Congratulations! Your manuscript is now with our production department. 

Kind regards, 

on behalf of

Dr. You-Yang Zhao 

Academic Editor

PLOS ONE